# The Maize Caterpillar *Mythimna* (= *Leucania*) *loreyi* (Duponchel, 1827) (Lepidoptera: Noctuidae): Identification, Distribution, Population Density and Damage in Tunisia

**DOI:** 10.3390/insects14100786

**Published:** 2023-09-26

**Authors:** Jouda Mediouni Ben Jemâa, Abir Soltani, Tasnim Djebbi, Ines Mejri, Dalton Kanyesigye, Michael Hilary Otim

**Affiliations:** 1Laboratory of Biotechnology Applied to Agriculture LR11INRAT06, National Agricultural Research Institute of Tunisia (INRAT), University of Carthage, Tunis 1004, Tunisia; soltany.abyr@gmail.com (A.S.); tasnimdjebbi1@gmail.com (T.D.); ines.mejri.tunisie@gmail.com (I.M.); 2National Crops Resources Research Institute (NaCRRI), Kampala P.O. Box 7084, Uganda; kanyesigyedalton@gmail.com (D.K.); motim9405@gmail.com (M.H.O.)

**Keywords:** *Mythimna loreyi*, maize, sorghum, identification, population dynamics

## Abstract

**Simple Summary:**

*Mythimna loreyi* Duponchel (Noctuidae) is reported on several gramineous hosts worldwide. It attacks particularly maize and reduces yields significantly. It is a native species in East Asia and found in the Near and Middle East, Australia and Africa. Recently, this insect emerged in Tunisia. It was detected abundantly during surveys carried out as part of a research project on the invasive insect pest *Spodoptera frugiperda* threatening maize, sorghum and other cereal crops in Africa. This work carried out the first investigations on its morphological and molecular identification, distribution, population density and damage in Tunisia. Surveys were carried out in four regions in the north and south of Tunisia where mostly maize and sorghum were planted (Jendouba, Bizerte, Nabeul, Gabes). Morphological characterization was based on male genitalia. Molecular identification was determined based on mitochondrial cytochrome oxidase I sequences (mtCOI). Field-collected data included adult monitoring using traps, infestations, and incidence of the pest. Correlations with whether parameters were studied. Overall, our results indicated that *M. loreyi* was well established in all prospected areas in Tunisia. Therefore, a control program is needed to manage this pest. Additionally, the threat of this insect on strategic cereal crops in Tunisia (wheat) should be further investigated.

**Abstract:**

Surveys were conducted during 2020 and 2021 to study the emerging lepidopteran pests inflicting cereals in Tunisia, with specific emphasis on maize and sorghum crops. A species was collected from traps placed in the Jendouba, Bizerte, Nabeul and Gabes regions. Thus, this study carried out first report on its identification, distribution, population density and damage. Results showed that *M. loreyi* was abundant in all prospected areas, with total adult captures reaching 4779 and 9499 moths on sorghum and maize, respectively, during 2020. Moreover, the mean infestation percentage reached its maximum during August at 31.05% and 20.69% for the Jendouba and Bizerte regions, respectively, while the highest infestations were observed in the Gabes and Nabeul regions during July, with respective mean values of 13.54% and 21.35%. In addition, results revealed that the highest pest incidence occurred in the Gabes region, with values of 11.1 ± 0.47 and 5.7 ± 0.48 during 2020 and 2021, respectively. Additionally, results pointed out that *M. loreyi* achieved two summer generations in the different localities of Tunisia. Overall, this study provides basic insights into the ecology and population biology of *M. loreyi*, which are required to establish an effective pest control program.

## 1. Introduction

The maize caterpillar *Mythimna* (= *Leucania*) *loreyi* (Duponchel, 1827) (Noctuidae) is commonly called the Loreyi leaf worm, cosmopolitan rice armyworm, rice cutworm, cereal armyworm, or false armyworm [1,2,3,4]. This noctuid is a native species in East Asia [5]. It undergoes multiple generations annually [6,7]. It is ranked among migratory Lepidopteran pests that reach the northern temperate zone [8]. This pest is reported on several gramineous hosts, including nine Poaceae species (*Arundo donax*, *Avena sativa*, *Oryza sativa*, *Pennisetum purpureum*, *Saccharum officinarum*, *Sorghum bicolor*, *Triticum aestivum*, *Triticum durum* and *Zea mays*), two Solanaceae species (*Capsicum* spp., *Nicotiana tabacum*) and one Fabaceae species (*Cicer arietinum)* [2,9,10,11]. *M. loreyi* has emerged in Africa, Australia, the Middle East and Asia [12].

Kornosor [13] reported that *M. loreyi* infests particularly maize and reduces yields significantly when attacks occurred just before silking and pollination. Likewise, Qian et al. [14] indicated that maize is the optimal host for *M. loreyi*, leading to the best biological performances (larval growth duration 20.18 days, survival rate egg-pupa 68%, female-pupa weight 32.4 mg). Earlier, Kornosor [13] cited that *M. loreyi* is a leaf feeder that invades maize from early growth stages (2–4 leaves) to pollen shedding and induced severe losses mainly on late-sown maize. Similarly, Sertkaya and Bayram [15] indicated that *M. loreyi* caused substantial economic damage to *Z. mays*. Additionally, Guo et al. [16] reported that *M. loreyi* usually occurred together with the closely related species *Mythimna separata* (Walker 1865) and caused considerable damage to host plants. Caterpillars bore into host plants and attacked the developing flower spikes. Moreover, Hirai et al. [17] cited that *M. loreyi* induced significant losses on crop production.

*Mythimna loreyi* attacks all parts of maize plants except the roots [18]. Newly hatched larvae feed on the tender leaves and make holes in their edges. As the larvae become older, they damage various parts of the plant as they feed on the developing tops and new growths of the leaves. They attack the stems and feed on grains in the milky stage. When they appear in large numbers, they cause economic damage to corn [19]. It is considered as one of important pests on the maize crop due to its attacks that reduce the yield [20], although there is little information available concerning *M. loreyi* population dynamics in the field [17].

In this regard, previous research indicated that climatic parameters such as temperature, relative humidity, rainfall, and wind speed significantly influenced the population dynamics of *M. loreyi* [18]. Furthermore, the insects’ development needs an optimum range, since any high or low temperature could diversely affect development, reproduction, and survival [21]. According to previous research conducted by Qin et al. [3], *M. loreyi* can develop, survive and lay eggs at temperatures ranging between 18 and 30 °C. Furthermore, temperature has greatest influence on insect multiple life-history variables, seasonality, insect activity and population dynamics in the field [3]. The adult insect most often comes out during the night period, especially at sunset. The adult is observed to emerge during the day and finds its way out of the cocoon through a letter-shaped slit. In the dorsal side of the cocoon, and immediately after its emergence, the adults move in different directions very slowly for about five minutes until their wings extend to their full size [16].

The number of *M. loreyi* generations varied from one country to another; for Korea, one generation has been observed against two generations in the Netherlands, North America, and Costa Rica [22]. In the central Mediterranean (Sicily), this noctuid moth occurs from March to November [23]. Additionally, Salis et al. [24] indicated that insect density influenced significantly damage to the crop. Also, these authors pointed out that changes in population densities and outbreaks of insects are mainly affected by biotic and abiotic constraints. Furthermore, Wang et al. [25] showed that increases in population density of *M. separata* caused greater damage to crops. Similarly, these authors demonstrated that study of insect densities could extend our knowledge of the agricultural pest’s biology and assist to develop appropriate management strategies.

Maize and sorghum are secondary cereal crops in Tunisia with limited planting areas, mainly used to ensure fodder production [26,27]. To the best of our knowledge, no previous studies have been undertaken on the entomofauna associated with *Z. mays* and *Sorghum* sp. in Tunisia. Additionally, the threat from this noctuid has become pronounced during the years 2020 and 2021 due to climate change, like increase in temperature. Consequently, there was an urgent need to identify this insect pest and investigate its population biology in different regions and various local climatic conditions. We focused in the present work on its morphological and molecular identification, distribution, population density and damage assessment. Moreover, we aimed to evaluate the influences of climatic parameters on seasonal fluctuation of *M. loreyi* and to highlight their potential importance in the improvement of pest management programs in Tunisia.

## 2. Materials and Methods

### 2.1. Experimental Fields

The survey for *M. loreyi* was conducted for two years, 2020 and 2021, in *Z. mays* crops in four regions of north and south Tunisia where maize is planted (Jendouba, Bizerte, Nabeul, Gabes). Two fields were chosen in each region, where no pesticides or chemical fertilization were applied.

Description of surveys’ localities of *M. loreyi* was reported in Table 1.

### 2.2. Weather Parameters

Meteorological data of each site were taken from the nearest meteorological station to the sampled fields (Figure 1). Mean monthly temperature, relative humidity and rainfall were recorded. The studied regions were characterized by a typical Mediterranean climate; summers are hot, muggy and dry, with cold winters. The average temperatures in these regions were 18.9, 18.8, 17.8 and 21.3 °C in 2020 against 18.75, 14, 14.1 and 21.3 °C in 2021 for Jendouba, Bizerte, Nabeul and Gabes, respectively. According to the Köppen climate classification, the climate sub-type for these areas is Csa (Mediterranean climate). For more details, Medhioub et al. [28] indicated that in the Jendouba region, the average temperature varied from 5 to 10 °C in winter and from 25 to 30 °C in summer. This region belongs to the humid climate, with an annual rainfall of to 1000 mm. Additionally, these authors cited that the Nabeul region is characterized by a temperate climate. The minimum temperature was 8.4 °C in January, while, in August, the temperature reached 22.6 °C.

The Gabes region, located in southeastern Tunisia, presented monthly average temperatures around 10 to 12 °C in winter against 30 °C in summer. Moreover, Jemai et al. [29] reported that the Gabes region had an arid Mediterranean climate, with mild wet winters and hot dry summers. Rainfall is typically scarce with annual mean varied between 100 and 200 mm [28].

For the Bizerte region, temperature varied between 7 °C as a minimum and 31 °C as a maximum during winter and summer, respectively. The annual average rainfall was between 510 and 638 mm [30].

### 2.3. Moth Collection and Identification

Since maize and sorghum are grown in Tunisia as summer crops, regular sampling was conducted during the period from June until September of each season. For immature insect stages, 30 leaves were weekly and randomly sampled from 30 host plants. Samples were taken to the laboratory and examined under a binocular magnifying glass to determine the number of eggs, larvae and pupae.

The adult population was monitored with plastic funnel traps baited with sex pheromone components as lures [31,32]. Moreover, trap density is considered one of the most important aspects for adult monitoring with respect to vegetation [33,34].

Adult moths were monitored using pheromone funnel traps, moth catchers or buckets (Unitrap^®^, Russel IPM^®^, Ltd., Flintshire, UK, Universal moth trap) suspended about 1.5 m above the ground and installed in the fields (Table 1), with one trap covering 0.5 ha at a density of 2 traps/ha [35]. Additionally, to ensure better trap catches, an insecticide was added to traps. Commercial lure of the fall armyworm (FAW) *Spodoptera frugiperda* (Smith 1797) sex pheromone (Russell IPM^®^, Ltd., Flintshire, UK) was employed and was replaced every 4 to 6 weeks depending on weather conditions. *S. frugiperda* sex pheromone lures were used as the initial objective of the surveys was to monitor the presence of this insect. In fact, this work is a part of a research project conducted through a consortium of many African countries and aims to develop an IPM program against the FAW for sustainable food security in Africa that was initiated after the invasion of this pest in Africa occurring first in 2016 in Nigeria. Catch records were taken weekly.

For temporal and geographical distribution, investigations were conducted in four regions, namely Jendouba, Bizerte, Nabeul and Gabes, presenting the most important areas of *Zea mays* and *Sorghum* sp. production in Tunisia.

### 2.4. Damage Assessment

Survey for *M. loreyi* infestation was conducted from early June until end of September. In each surveyed field, 50 plants were collected at random by “W” sampling method (sampling across the field at 10–15 m) [36]. The number of infested plants and plant damage were recorded. Foliar damage was assessed based on a visual scale ranging from 0 to 5 scores as described: 0 = plant with no visual foliar damage; 1 = up to 10% of foliar damage; 2 = foliar damage between 10 and 25%; 3 = foliar damage between 25 and 50%; 4 = foliar damage between 50 and 75%; 5 = more than 75% of foliar damage or a dead plant. Field surveys were carried out during the daylight period, from 7 h to 17 h. Given that the pupal stage of *M. loreyi* normally occurs in the soil, this stage was deliberately excluded from the survey. Moreover, sprayed fields were excluded from the survey.

### 2.5. Pest Incidence

Samples were carefully examined to determine the total number of infested leaves and the number of pre-imaginal stages on each infested leaf [37].
(1)$$\mathrm{Pest}\;\mathrm{incidence}=\frac{\mathrm{Number}\;\mathrm{of}\;\mathrm{preimaginal}\;\mathrm{stages}}{\mathrm{Number}\;\mathrm{of}\;\mathrm{infested}\;\mathrm{leaves}}$$

### 2.6. Morphological Identification

To identify the specimens, the abdomen was boiled in 10% KOH for approximately ten minutes. Then, the macerated abdomen was kept in 70% alcohol. After that, the genitalia were stained in 1% chlorazol black which was dissolved in 30% alcohol and cleared in 70% alcohol. Finally, specimens were identified according to Rungs [38].

### 2.7. Molecular Identification

#### 2.7.1. DNA Extraction

The DNA was extracted from five samples. DNA from each individual sample (leg) was isolated using the chelex 100 resin method as described by Otim et al. [39]. Before extraction, each sample (adult) was washed in sterile distilled water, and a new sterile surgical blade was used to cut off a leg, and the remaining part of the sample was returned to the tube for storage. Each cut leg was placed in a sterile 1.5 mL Eppendorf tube, and 50 μL of 10% chelex 100 solution (BioRad) was added, followed by 10 μL (20 mg/mL) of proteinase K solution (Bioline). The samples were incubated at 56 °C overnight, followed by brief vortex and heat inactivation at 100 °C for 15 min. The resulting mixture was centrifuged at 15,900 relative centrifugal forces for 3 min, and then 40 μL supernatant was collected in a new sterile 1.5 mL Eppendorf tube and stored at −20 °C.

#### 2.7.2. Polymerase Chain Reaction (PCR)

Polymerase chain reaction of all the moth adult samples was carried out as explained by Otim et al. [40] using primers (LCO1490: 5′-GGTCAACAAATCATAAAGATATTGG-3′; HC02198: 5′-TAAACTTCAGGGTGACCAAAAAATCA-3′ [40].

The mitochondrial DNA cytochrome c oxidase subunit 1 (mtCOI) gene of all the samples was amplified by polymerase chain reaction (PCR) using the LCO1490 and HCO2198 primer set in a 25 μL volume per reaction. The PCR mixture consisted of 16.25 μL of nuclease-free water (Thermo Fisher Scientific Inc., Waltham, MA, USA), 2.5 μL of 10× Dream Taq green buffer, 0.5 μL deoxynucleotide triphosphate (dNPTs) (10 mM) (Thermo Fisher Scientific Inc.), 1 μL of each primer (10 Pmol/μL), 0.25 μL of Dream Taq DNA polymerase (Thermo Fisher Scientific Inc.), 2.5 μL of 5% Tween 20 (Sigma Aldrich, Darmstadt, Germany) and 1 μL of the template DNA. The PCR thermocycling conditions were performed in a Biometra thermocycler (Analytik Jena GmbH, Jena, Germany) as follows: 94 °C (2 min) for initial denaturation, followed by 35 cycles of 94 °C (30 s), 52 °C (35 s) and 72 °C (45 s) and a final extension step at 72 °C for 10 min. Post PCR reactions were stored at 4 °C until time for sequencing. PCR products were visualized by electrophoresis in 1× TAE buffer 1.5% (*w*/*v*) agarose gels (UltraPure Agarose, Invitrogen, Carlsbad, CA, USA) stained with 5 μL of ethidium bromide, visualized under UV light to confirm the presence of the amplicon and photographed using a digital camera in U: GENIUS3 gel documentation system (Syngene, Cambridge, UK).

#### 2.7.3. Sequence Analysis

After sequencing, the sequence trace files were processed using pregap 4 and gap 4 [41]. After processing, the short and poor sequences were discarded. The clean sequences were then compared with the already existing sequences in the NCBI gene bank via the basic local alignment search tool (BLAST) [42]. The sequences were checked for premature stop codons using Geneious prime (https://www.geneious.com, accessed on 8 July 2022). The sequences were then aligned in MEGA 7 [43] and the same software was used for the maximum-likelihood phylogenetic tree inference and generating the pairwise distances using 1000 bootstraps.

### 2.8. Statistical Analysis

Statistical analyses were performed in order to understand data using the SPSS statistical software version “20” [44]. For eggs, larvae, pupae and captured adults, data were expressed as mean ± SD. Data were analyzed using analysis of variance with Fisher test. The mean infestation rate and pest incidence was calculated for two years 2020 and 2021 for each month and were compared at *p* ≤ 0.05 using Duncan’s test. Comparisons were made between regions and crop seasons using one-way ANOVA followed by the Duncan test. Sampling data of each developmental stage of *M. loreyi* during the two seasons 2020 and 2021 were analyzed using generalized linear models (GLM). When necessary, data were transformed by common logarithm to meet the assumption of normality. To establish the relationships between weather parameters and *M. loreyi* flight activity, Bravais-Pearson «r» coefficient was calculated.

## 3. Results

### 3.1. Morphological Identification

Based on the description given by Rishi and Khokhar [45], morphological identification revealed that this species belongs to the order Lepidoptera, Noctuidae family and the genus *Mythimna*: the Female body (♀) measured about 14 mm (Figure 2A) against 16.3 mm for the male adult (♂) (Figure 2B).

This species was characterized by spherical head in shape and somewhat depressed dorso-ventrally with its sides rounded in both male and female (Figure 2A,B). The antennae are multi-segmented (Figure 2C). This species presented brown wings with pale-brown veins (Figure 2D,E). In addition, legs are also brown (Figure 2F).

The examination of the male genitalia revealed a long, curved spine which prolongs the saccus (Figure 3) that characterized *M. loreyi* according to Rungs [38].

### 3.2. Molecular Identification

Only the samples that amplified successfully and showed a band of the expected size (710 bp) were sequenced. Maximum-likelihood inference resulted in our samples (5 and 7) (Figure 4). Table 2 reports the pairwise distances of our samples and other species downloaded from NCBI. Results revealed clustering with *M. loreyi* with very close genetic distances.

### 3.3. Damage and Pest Incidence

Infestation percentage was evaluated during plant growth. The infestation was determined based on the total number of infested leaves per total number of collected leaves.

In 2020, the infestation reached its maximum during August with mean percentages of 31.05% and 20.69% for Jendouba and Bizerte, respectively, while the highest infestations were observed in the Gabes and Nabeul regions during July (Table 3). In 2021, the mean infestation percentages were 24.2% and 18.41%, respectively, during August for Jendouba and Bizerte and during July for Gabes and Nabeul fields, respectively. For Jendouba fields (the region with largest areas planted with maize and sorghum), infestation rates were 22.58 ± 0.1 and 18.24 ± 0.23 during 2020 and 2021, respectively.

The incidence of *M. loreyi* was determined based on the mean number of the immature stage (eggs, larvae and pupae). For the assessment of *M. loreyi* incidence, results indicated that Jendouba was the region where the incidence of the insect was the most important (Table 3). Moreover, August and July were the months with the highest incidence of *M. loreyi* for 2020 and 2021, respectively.

### 3.4. Population Density

Figure 4 illustrates the population density of *M. loreyi* for the four regions during 2020 and 2021. *M. loreyi* population density was expressed as numbers of captured adults as well as numbers of pre-imaginal stages (eggs, larvae and pupae) along the cycles of host crops. Results showed that flight activity depends on different parameters such as region and capture period. Moreover, the highest numbers of captured adults have been observed during July and August on maize, with captures reaching 1383 moths during 2020 for the Jendouba region against 3681 moths during 2021. However, the results reported in Figure 4 demonstrate that flight activity of moths reached its maximum during July for the Bizerte, Gabes and Nabeul regions. Likewise, the results revealed the presence of two adult peaks corresponding probably to the two annual summer generations of this insect. The first generation began in June or July depending on the region and year (Figure 4). The second generation was observed during mid-August with 980 and 2158 moths for Jendouba and 57 and 58 moths for Bizerte. However, for Gabes, the second generation was observed during the first week of September, and for the Nabeul region, the second generation was observed during the second week of July (Figure 4). The obtained data demonstrate that *M. loreyi* was first observed at the beginning of June, which coincides with *Zea mays* vegetative stage. The number of captured adults increased as the crop developed, reaching its maximum in July (corn cob formation stage). The number of emerging adults varied significantly with locality (df = 3, F = 201.5, *p* ≤ 0.01). In addition, statistical significant differences were observed between the two years (df = 1, F = 3.6, *p* ≤ 0.05).

From June until September (the period during adults flying), the females laid their eggs on the maize leaves. During July and August, the egg numbers were higher than those registered during June and September for the Jendouba and Bizerte regions. Though for the Nabeul and Gabes regions, the highest number of eggs was observed during June.

Larval activity was monitored during July and August. A significant increase in the larval population was observed during the first two weeks of August (Figure 5). The first generation lasted for more than 30 d, while the second generation lasted 26 d (Figure 5).

### 3.5. Temporal and Geographical Distribution of M. loreyi

The total monthly number of adult moths captured in the four regions of Tunisia from June to September during 2020 and 2021 is illustrated on Figure 5 (*Zea mays)* and Figure 6
*(Zea mays* and *Sorghum* sp.). The results revealed that *M. loreyi* occurred on both crops and that the captures depended on the region and the crop. The total numbers of captured adults were 100, 1578, 91 and 29 on maize during 2020 and 1220, 24, 26 and 25 during 2021, respectively, for the Jendouba, Bizerte, Gabes and Nabeul regions against 1112, 3126, 61 and 62 during 2020 and 237, 0, 4 and 0 during 2021 on sorghum for Jendouba, Bizerte, Gabes and Nabeul.

Statistical analysis revealed high significant differences between regions in relation to total numbers of captured adults (df = 5, F = 1,033,601.49, *p* ≤ 0.00). Moreover, significant differences have been detected between crops (df = 5, F = 130,308.27, *p* ≤ 0.00). Additionally, the interaction between region X crops showed significant differences (df = 25, F = 868,064.78, *p* ≤ 0.001).

### 3.6. Correlation between Pest Incidence and Weather Parameters

Table 4 shows the correlations between the number of monthly captured adults of *M. loreyi* and climatic parameters such as mean temperature, humidity and rainfall.

Results revealed significant and negative correlations between mean temperature and captured adults (r = −0.287, *p* ≤ 0.05). However, no significant correlations have been observed between captured adults and humidity and rainfall (Table 4).

## 4. Discussion

Our phylogenetic analysis of mt*COI* partial gene revealed that our samples belonged to *M. loreyi* with very close genetic distances, and the molecular identification found the same result as morphological identification. This study has confirmed the existence of *M. loreyi* in Tunisia via a combination of morphological identification and molecular barcoding methods.

Usually, insect population dynamics are subjected to various environmental factors, like temperature [46], which is considered a major factor influencing *M. loreyi* emergence [3]. Our work in the population biology of *M. loreyi* pointed out the presence of two summer generations in different localities of Tunisia. The highest densities were registered in July that coincides with flowering stage of host plants. Additionally, the highest number of captured adults was registered for the second generation. However, the lowest densities were observed during June. The number of captured adults varied according to climatic parameters, such as the mean temperature, that occurred during July 2020 and 2021. Additionally, significant negative correlation was observed between the number of captured adults and the mean temperature values, at *p* ≤ 0.05. Thus, the high temperature that occurred during season crops (July and August) influenced negatively *M. loreyi* emergence. Our research is in accordance with previous work, which noted that insect pest population dynamics were highly dependent on climatic parameters [47,48]. Similarly, other research, reported by Gilbert and Raworth [49] and Kiritani [50], indicated the high influence of temperature on insect biology. Currently, *M. loreyi* is becoming one of the most serious grain pests in Africa. *M. loreyi* presents multiple generations per year [45]. In the same context, Qin et al. [3] revealed that *M. loreyi* populations could complete two generations at 30 °C. Moreover, obtained data showed significant differences on the number of captured adults between years and between months, which could be related to climatic conditions. These results can be supported by other studies that revealed the significant effect of abiotic factors like temperature on the life cycle, emergence and oviposition of insects [51]. Likewise, the results of this study recommend that the use of pheromone funnel traps could be considered as an efficient tool for the capture of *M. loreyi* under field conditions. Additionally, the outcomes of other research conducted by Hong et al. [22] indicated that climatic parameters influenced the release rate of pheromones and therefore affect the population dynamics of *M. loreyi.*

Noctuid species are considered among the most dangerous pests to maize [52]. Our results revealed that *M. loreyi* caused substantial damage, with a 31.05% infestation rate during 2020 in the Jendouba region. In this regard, previous works pointed out the importance of pest damage on crop losses and yield reduction. Indeed, it was demonstrated that 100% of crop losses occurred if control measures were not taken at suitable times [53]. Furthermore, results showed that infestation caused by *M. loreyi* varied through the time and differed from one region to another. The lowest infestation and pest incidence were recorded in the Gabes region during September. These results could be discussed by other works, which demonstrated that adverse climatic parameters that insects encounter in the field are usually transient and could motivate insects to migrate to other regions [12,54].

On the other hand, results demonstrated that *M. loreyi* attacked both crops, maize and sorghum, and that the captures depended on the region and the crop. Similarly, Kara et al. [55] reported that *M. loreyi* is a severe pest of gramineous crops that specifically reduced yields in maize plants. However, total numbers of captured adults in sorghum fields were less compared to the captured adults on maize. In this context, Singh and Chaudhary [56] indicated that *M. loreyi* species had moderate performance on sorghum crops.

This study revealed that *M. loreyi* is well distributed in Tunisia and has a seasonal abundance that depends on host plants and regions. Moreover, this research documented the impact of temperature on adult captures and consequently on population dynamics. Thus, ecological, environmental and phenological factors must be further investigated to manage this pest population.

## Figures and Tables

**Figure 1 insects-14-00786-f001:**
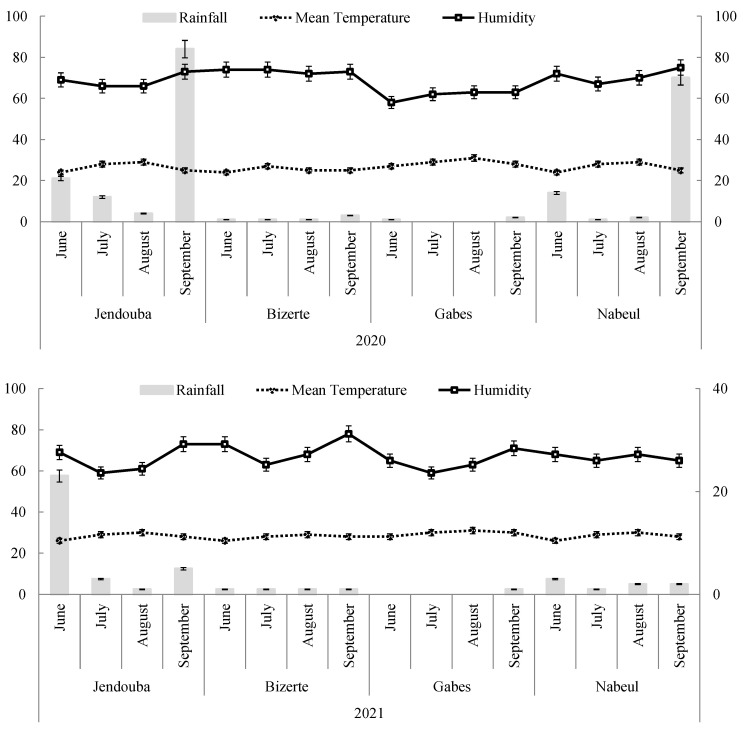
Climatic parameters of Jendouba, Bizerte, Gabes and Nabeul during 2020 and during 2021.

**Figure 2 insects-14-00786-f002:**
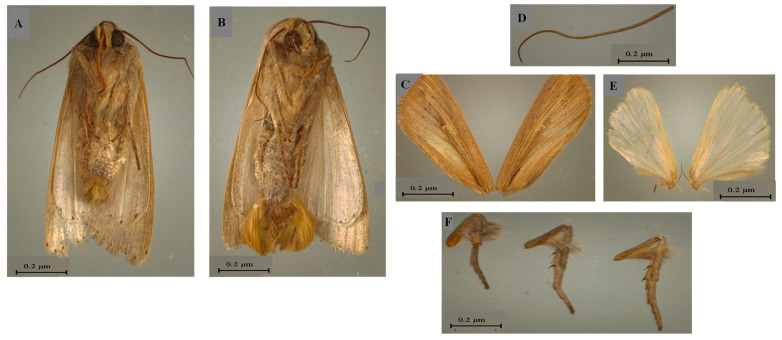
Adults. (**A**) Female of *Mythimna loreyi*. (**B**) Male *of Mythimna loreyi*. (**C**) Antenna of male. (**D**) Anterior wings are pale-brown with small dark spot in the middle. (**E**) Posterior wings. (**F**) Legs of male.

**Figure 3 insects-14-00786-f003:**
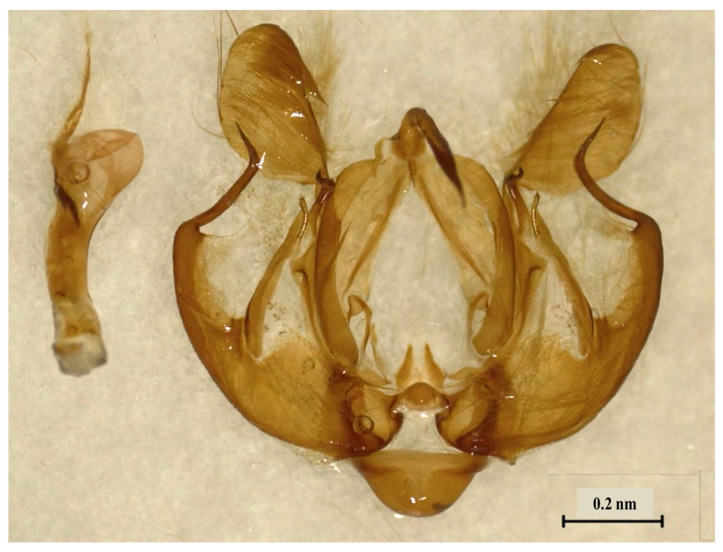
Male genitalia of *Mythimna loreyi* collected from Tunisia (Jendouba region) during 2021 season.

**Figure 4 insects-14-00786-f004:**
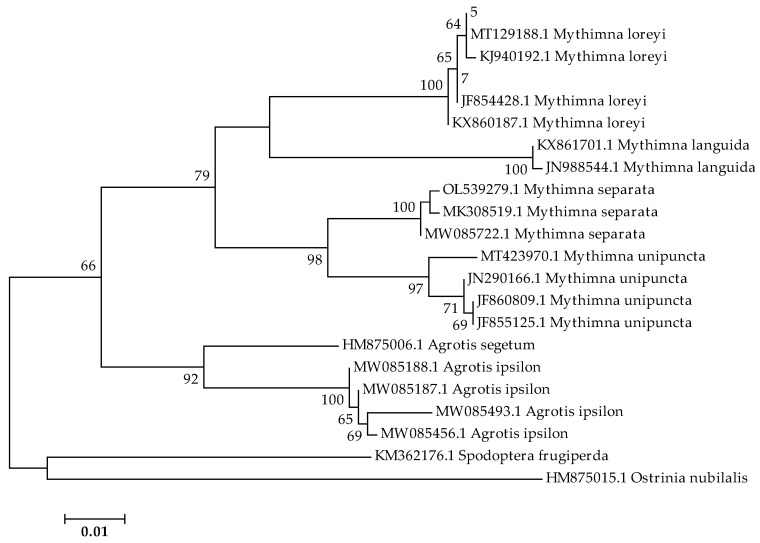
Maximum-likelihood phylogenetic tree inferred in MEGA 7 using 1000 bootstraps, showing the genetic relationships of our samples (5 and 7) and the sequences of other species downloaded from NCBI.

**Figure 5 insects-14-00786-f005:**
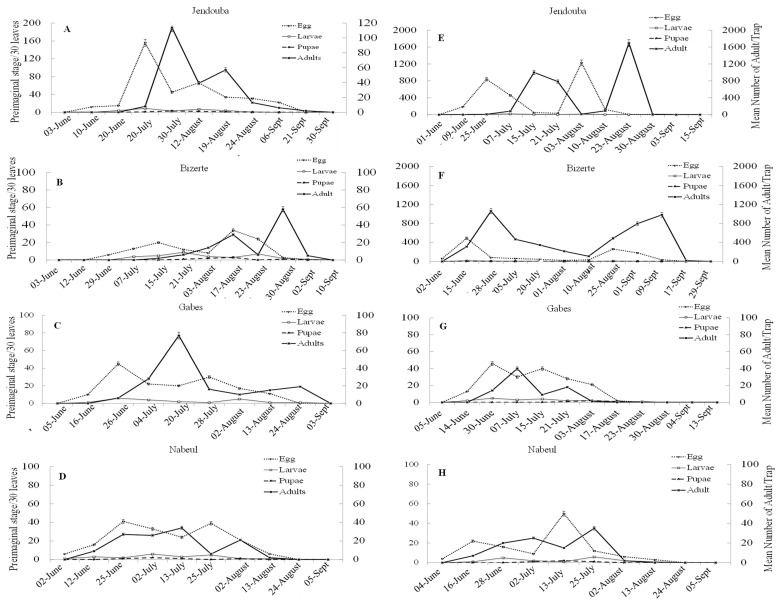
(**A**–**H**) Mean number of captured *Mythimna loreyi* adults in pheromone traps and preimaginal stages on maize leaves in different regions of Tunisia during 2020 and 2021.

**Figure 6 insects-14-00786-f006:**
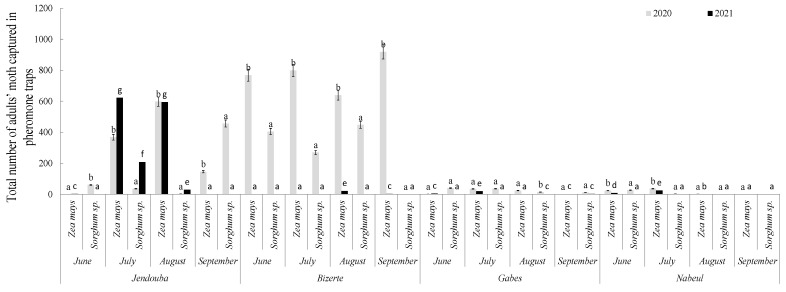
Total number of adult moths captured in pheromone traps in different regions of Tunisia during 2020 and 2021 on *Zea mays* and *Sorghum* sp. Different letters a, b and c indicate significant differences at (*p* ≤ 0.001) between regions according to Duncan test.

**Table 1 insects-14-00786-t001:** Description of fields sampled in each region.

Region	Site	Altitude (m)	Area (Ha)	Number of Traps
Jendouba	El Marja	129	60	120
	Bir Lakhder	126	42	84
Bizerte	Mateur	13	6	12
	Utique	14	6	12
Gabes	Chnenni	380	3	6
	Taboulbou	15	3	6
Nabeul	Manzel Tmim	15	1.5	3
	Manzel Tmim	19	1.5	3

**Table 2 insects-14-00786-t002:** The pairwise distances of our samples and other species downloaded from NCBI. The distances are shown below the diagonal, and the standard errors are shown above the diagonal.

5		0.001	0.001	0.002	0.001	0.000	0.013	0.010	0.011	0.011	0.011	0.011	0.011	0.011	0.011	0.011	0.016	0.012	0.013	0.013	0.012	0.013
7	0.002		0.000	0.001	0.002	0.001	0.013	0.010	0.011	0.011	0.011	0.011	0.011	0.011	0.011	0.010	0.016	0.012	0.013	0.012	0.012	0.012
JF854428.1_*Mythimna_loreyi*	0.002	0.000		0.001	0.002	0.001	0.013	0.010	0.011	0.011	0.011	0.011	0.011	0.011	0.011	0.010	0.016	0.012	0.013	0.012	0.012	0.012
KX860187.1_*Mythimna_loreyi*	0.003	0.002	0.002		0.003	0.002	0.013	0.010	0.011	0.011	0.011	0.011	0.011	0.010	0.011	0.010	0.015	0.012	0.013	0.012	0.012	0.012
KJ940192.1_*Mythimna_loreyi*	0.002	0.003	0.003	0.005		0.001	0.013	0.011	0.011	0.011	0.011	0.011	0.012	0.011	0.011	0.011	0.016	0.012	0.013	0.013	0.013	0.013
MT129188.1_*Mythimna_loreyi*	0.000	0.002	0.002	0.003	0.002		0.013	0.010	0.011	0.011	0.011	0.011	0.011	0.011	0.011	0.011	0.016	0.012	0.013	0.013	0.012	0.013
KM362176.1_*Spodoptera_frugiperda*	0.102	0.101	0.101	0.099	0.104	0.102		0.013	0.013	0.013	0.013	0.014	0.014	0.014	0.014	0.014	0.015	0.014	0.014	0.014	0.014	0.014
MT423970.1_*Mythimna_unipuncta*	0.068	0.067	0.067	0.065	0.070	0.068	0.099		0.004	0.005	0.005	0.010	0.011	0.008	0.008	0.008	0.016	0.011	0.013	0.012	0.012	0.012
JN290166.1_*Mythimna_unipuncta*	0.072	0.070	0.070	0.072	0.074	0.072	0.099	0.014		0.001	0.001	0.010	0.011	0.008	0.008	0.008	0.017	0.012	0.013	0.013	0.013	0.013
JF860809.1_*Mythimna_unipuncta*	0.074	0.072	0.072	0.074	0.075	0.074	0.099	0.015	0.002		0.000	0.010	0.011	0.007	0.008	0.007	0.017	0.012	0.013	0.012	0.012	0.012
JF855125.1_*Mythimna_unipuncta*	0.074	0.072	0.072	0.074	0.075	0.074	0.099	0.015	0.002	0.000		0.010	0.011	0.007	0.008	0.007	0.017	0.012	0.013	0.012	0.012	0.012
KX861701.1_*Mythimna_languida*	0.077	0.075	0.075	0.074	0.079	0.077	0.109	0.075	0.075	0.074	0.074		0.002	0.011	0.011	0.011	0.016	0.012	0.013	0.013	0.013	0.013
JN988544.1_*Mythimna_languida*	0.079	0.077	0.077	0.075	0.081	0.079	0.111	0.077	0.077	0.075	0.075	0.002		0.011	0.011	0.011	0.016	0.013	0.014	0.013	0.013	0.013
OL539279.1_*Mythimna_separata*	0.068	0.067	0.067	0.065	0.070	0.068	0.104	0.039	0.036	0.034	0.034	0.077	0.079		0.002	0.002	0.016	0.012	0.013	0.012	0.012	0.012
MK308519.1_*Mythimna_separata*	0.072	0.070	0.070	0.068	0.074	0.072	0.108	0.042	0.039	0.038	0.038	0.077	0.079	0.003		0.002	0.016	0.012	0.013	0.012	0.012	0.012
MW085722.1_*Mythimna_separata*	0.068	0.067	0.067	0.065	0.070	0.068	0.104	0.039	0.036	0.034	0.034	0.073	0.075	0.003	0.003		0.016	0.012	0.013	0.012	0.012	0.012
HM875015.1_*Ostrinia_nubilalis*	0.142	0.140	0.140	0.138	0.144	0.142	0.136	0.151	0.149	0.149	0.149	0.151	0.153	0.140	0.144	0.140		0.014	0.016	0.015	0.015	0.015
HM875006.1_*Agrotis_segetum*	0.087	0.085	0.085	0.087	0.087	0.087	0.111	0.077	0.082	0.080	0.080	0.091	0.092	0.085	0.089	0.085	0.122		0.010	0.009	0.008	0.008
MW085493.1_*Agrotis_ipsilon*	0.104	0.102	0.102	0.101	0.106	0.104	0.113	0.096	0.101	0.099	0.099	0.104	0.106	0.092	0.096	0.092	0.145	0.062		0.004	0.005	0.004
MW085456.1_*Agrotis_ipsilon*	0.096	0.094	0.094	0.092	0.097	0.096	0.108	0.087	0.092	0.090	0.090	0.094	0.096	0.084	0.087	0.084	0.134	0.052	0.012		0.003	0.002
MW085188.1_*Agrotis_ipsilon*	0.094	0.092	0.092	0.090	0.096	0.094	0.104	0.085	0.090	0.089	0.089	0.092	0.094	0.082	0.085	0.082	0.129	0.047	0.014	0.005		0.002
MW085187.1_*Agrotis_ipsilon*	0.096	0.094	0.094	0.092	0.097	0.096	0.106	0.087	0.092	0.090	0.090	0.094	0.096	0.084	0.087	0.084	0.131	0.049	0.012	0.003	0.002	

**Table 3 insects-14-00786-t003:** Infestation (%) and incidence of *M. loreyi* during 2020 and 2021 on *Zea mays* in Jendouba, Bizerte, Nabeul and Gabes regions (highest values were marked in **bold**).

		Infestation (%)		Incidence	
Locality	Month	2020	2021	2020	2021
Jendouba	June	24.91 ± 0.45	16.59 ± 0.95	3.3 ± 0.51	2.6 ± 0.26
	July	25.21 ± 0.18	20.13 ± 0.13	6.3 ± 0.54	**5.0 ± 0.45**
	August	**31.05 ± 0.24**	**24.2 ± 0.84**	**7.3 ± 0.10**	3.6 ± 0.84
	September	22.58 ± 0.1	18.24 ± 0.23	3.5 ± 0.23	0.0 ± 0.0
Bizerte	June	11.23 ± 0.14	9.2 ± 0.1	1.2 ± 0.56	0.0 ± 0.00
	July	18.22 ± 0.54	17.21 ± 0.21	2.2 ± 0.01	**3.5 ± 0.12**
	August	**20.7 ± 0.84**	**18.41 ± 0.65**	**2.9 ± 0.01**	1.9 ± 0.1
	September	17.24 ± 0.59	15.34 ± 0.45	0.0 ± 00.00	0.2 ± 0.01
Nabeul	June	14.12 ± 0.72	10.33 ± 0.84	1.4 ± 0.02	1.9 ± 0.02
	July	**21.35 ± 0.46**	**13.41 ± 0.48**	2.2 ± 0.01	**2.2 ± 0.01**
	August	20.32 ± 0.13	9.60 ± 0.50	**2.3 ± 0.01**	1.0 ± 0.01
	September	2.22 ± 0.02	0.98 ± 0.03	0.0 ± 0.00	0.0 ± 0.00
Gabes	June	2.1 ± 0.45	0.9 ± 0.01	**11.1 ± 0.47**	2.9 ± 0.32
	July	**13.54 ± 0.23**	**5.3 ± 0.03**	1.7 ± 0.12	**5.7 ± 0.48**
	August	10.87 ± 0.41	3.21 ± 0.01	4.2 ± 0.26	2.1 ± 0.02
	September	2.1 ± 0.02	0.1 ± 0.01	0.0 ± 0.0	0.0 ± 0.0

**Table 4 insects-14-00786-t004:** Correlations between number of captured adults and weather parameters (mean temperature, humidity and rainfall).

	Month	Captured Adults	Mean Temperature	Humidity	Rainfall	Infestation (%)
Month	1	0.012	0.287 *	0.264 *	0.226	0.032
Captured adults	-	1	−0.248 *	0.142	−0.118	0.682 **
Mean Temperature	-	-	1	−0.497 **	−0.452 **	−0.521 **
Humidity	-	-	-	1	0.303 *	0.275
Rainfall	-	-	-	-	1	−0.116
						1

** Correlation is significant at the 0.01 level (2-tailed). * Correlation is significant at the 0.05 level (2-tailed).

## Data Availability

Data will be available when needed.

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
