# Peer review of "The Maize Caterpillar Mythimna (= Leucania) loreyi (Duponchel, 1827) (Lepidoptera: Noctuidae): Identification, Distribution, Population Density and Damage in Tunisia"

_insects, 2023, doi:10.3390/insects14100786_

Round 1

Reviewer 1 Report (Previous Reviewer 1)

To the attention of the Authors:

some corrections and comments to be made on the text

Author Response

ANSWER LETTER

Dear Editor,

We would like to thank you and the reviewers for the helpful feedback and verdict on our manuscript. In response to the comments of reviewers we have made the suggested modifications and highlighted them in yellow color in the text. Detailed corrections are listed point by point herein.

Reviewer 1#:

1-change in: (Duponchel, 1827)

Thank you for your comment, the recommended modification has been done

2- larval

Thank you for your comment, the recommended modification has been done

3-The following contribution has not been added, as previously suggested. In references, number 23 reports other authors! Russo, P.; Bella, S.; Parenzan, P. Contributo alla conoscenza dei Nottuidi della Sicilia (Lepidoptera, Noctuidae). Phytophaga 2002, XI, 11-85.

Thank you for your comment, the reference has been added as you recommended.

4- non italic e add space: sp. In

Thank you for your comment, the recommended modification has been done

5- delete adult! Is there a "non-adult" female?

Thank you for your comment, the recommended modification has been done

6- the wingspan of M. loreyi is about 34-44 mm. So, is 14 or 16.3 mm the length of just one wing? Or that of the body?

Thank you for your comment, the values of 14 or 16.3 presented the length of females and males’ bodies respectively.

7- delete adult! Is there a "non-adult" male?

Thank you for your comment, the recommended modification has been done

8- replace "for male and female adults" with "in both male and female"

Thank you for your comment, the recommended modification has been done

9- change to: Figure 2. Adults. (A) Female of Mythimna loreyi, (B) Male of Mythimna loreyi, (C) The Antenna of male, (D)...

Thank you for your comment, the recommended modification has been done

10- Legs of the male or female?

Thank you for your comment. “Leg of male”

11- genitalia

Thank you for your comment, the recommended modification has been done

Reviewer 2 Report (Previous Reviewer 2)

I've recently reviewed the work of Ben Jemaa and colleagues entitled "The maize caterpillar Mythimna (= Leucania) loreyi (Duponchel. 1827) (Lepidoptera: Noctuidae): Identification, distribution, population density and damage in Tunisia". The authors considered suggestions and tried to correct the manuscript accordingly. However, as I wrote in the previous revision I made, the work doesn't sound as a "original" scientific paper. The authors just reported data coming from a monitoring program that, on the basis of M&Ms, was conducted towards Spodopotera litoralis. The authors did not report results from a region-scale monitoring program but included only data from a farm-scale monitoring program. From this point of view, entitling the work as "...distribution, population density in Tunisia" is quite speculative.  Moreover, no results about eggs and larvae are reported in the text, which could be more interesting for reader than morphological/genetic identification. I'm really sorry, but I believe the work is not yet acceptable for publication in Insects.

No specific comment

Author Response

ANSWER LETTER

Dear Editor,

We would like to thank you and the reviewers for the helpful feedback and verdict on our manuscript. In response to the comments of reviewers we have made the suggested modifications and highlighted them in yellow color in the text. Detailed corrections are listed point by point herein.

Reviewer 2#:

I've recently reviewed the work of Ben Jemaa and colleagues entitled "The maize caterpillar Mythimna (= Leucania) loreyi (Duponchel. 1827) (Lepidoptera: Noctuidae): Identification, distribution, population density and damage in Tunisia". The authors considered suggestions and tried to correct the manuscript accordingly. However, as I wrote in the previous revision I made, the work doesn't sound as a "original" scientific paper. The authors just reported data coming from a monitoring program that, on the basis of M&Ms, was conducted towards Spodopotera litoralis. The authors did not report results from a region-scale monitoring program but included only data from a farm-scale monitoring program. From this point of view, entitling the work as "...distribution, population density in Tunisia" is quite speculative.  Moreover, no results about eggs and larvae are reported in the text, which could be more interesting for reader than morphological/genetic identification. I'm really sorry, but I believe the work is not yet acceptable for publication in Insects.

Response : Thank you for your comments and helpful critics. In this study, we presented data of the two years surveys conducted only in four regions of Tunisia namely Nabeul, Bizerte, Jendouba and Gabes. It is true that is not representative of the whole region, but these areas are the most important one in terms of cereal production in Tunisia. The work is still ongoing and other sites were added. It is a part of a research project conducted through a consortium of 15 African countries that aims to develop an IPM program against the Fall Armyworm (Spodoptera frugiperda) for sustainable food security in Africa. It is a TCP supported by Korea-Africa for Food and Agriculture Cooperation Initiative (KAFACI). This TCP was initiated after the invasion of the FAW to Africa occurring first in 2016 in Nigeria. It is interesting to mentioned that Tunisia is still free from S. frugiperda, but the survey showed the occurrence of M. loreyi.

Comments on the Quality of English Language : No specific comment

Response : Thank you for your comments

Reviewer 3 Report (Previous Reviewer 3)

The revised version of the manuscript has been improved by adding my comments related with abstract, results and Discussion. Significant changes have been done in the revised version.

Miner revision needed

Author Response

ANSWER LETTER

Dear Editor,

We would like to thank you and the reviewers for the helpful feedback and verdict on our manuscript.

Reviewer 3#:

Comments for Authors

The revised version of the manuscript has been improved by adding my comments related with abstract, results and Discussion. Significant changes have been done in the revised version.

Abstract

The Authors have added the in numerical values related with distribution, population density and  damage in Tunisia from.

-Thank you for your comment.

Results

The authors have amended this part by adding the missing figures of male and female both. 

Discussion

The Discussion has been extended. 

- Thank you for your comment.

General Comments.

Now the manuscript is has been significantly improved by authors. All suggestions have been technically amended.

- Thank you for your comment.

This manuscript is a resubmission of an earlier submission. The following is a list of the peer review reports and author responses from that submission.

Round 1

Author Response

ANSWER LETTER

Dear Editor,

We would like to thank you and the reviewers for the helpful feedback and verdict on our manuscript. In response to the comments of reviewers we have made the suggested modifications and highlighted them in red color in the text. Detailed corrections are listed point by point herein.

Reviewer 3#:

1-delete space

Response 1: Thank you for your comment, the recommended modification has been done

2-loreyi (Duponchel)

Response 2: Thank you for your comment, the recommended modification has been added

 3-2022 or 2023?

Response 3: Thank you for your comment, “Insects 2023, 13, x FOR PEER REVIEW”

 4- durum?

Response 4: Thank you for your comment, the recommended modification has been done “Triticum durum”

 5- uniforms the coordinates

Response 5: Thank you for your comment, the coordinates was deleted as it is recommended by other reviewer.

 6- in italics and author's name

Response 6: Thank you for your comment, done. “Spodoptera frugiperda (Smith 1797)»

 7- reduce space

Response 7: Thank you, done.

 8- Lepidoptera;

Response 8: Thank you for your comment, the recommended modification has been done “Lepidoptera,”

 9- in italic

 Response 9: Thank you for your comment, the recommended modification has been done “Mythimna”

 10- only three segments? this is incorrect or poorly written! Spodoptera littoralis, for example, has around 80 segments..... (Seada, 2015)

Response 10: Thank you for your comment, the recommended modification has been done. “The antenna presented different segments»

 11- add space

Response 11: Thank you for your comment, the recommended modification has been done “2021 on sorghum”

 12-delete spaces

Response 12: Thank you for your comment, the recommended modification has been done “(df=5. F=1033601.49. P≤0.00). Moreover, significant differences have been observed between crops (df=5. F=130308.27. P≤0.00). Additionally, interaction between region X crops showed significant differences (df=25. F=868064.78. P≤0.001). »

 13-delete the two photos and replace with a photo of a specimen in good condition: the specimen is so damaged that the species is not recognizable! Furthermore, no specific morphological characteristics

Response 13: Thank you for your comment, the recommended modification has been done

 14-wrong, this is a ventral view

Response 14: Thank you for your comment, It is corrected.

 15-rotate the figure, place the uncus in the upper position

It is male or female...?

add also locality and date

Response 15: Thank you for your comment, the recommended modification has been done

 16-preimaginal?

Response 16: Thank you for your comment, the figure has been corrected.

 17-no space

Response 17: Thank you for your comment, done. “In addition, significant negative correlation was observed between number of captured adults and mean temperature at P≤0.05”

 18- work

Response 18: Thank you for your comment, the recommended modification has been corrected.” with previous work which noted”

 19- loreyi

Response 19: Thank you for your comment, the recommended modification has been done “Leucania loreyi”

 20- Solanum sisymbriifolium (In Italic)

Response 20: Thank you for your comment, the recommended modification has been done

 21- Florida

 Response 21: Thank you for your comment, the recommended modification has been done “with pheromones in Mexico, Florida. Entomol»

 22- add space

Response 22: Thank you for your comment, the recommended modification has been done

 23- One

Response 23: Thank you for your comment, the recommended modification has been done

24- temperature - A general

 Response 24: Thank you for your comment, the recommended modification has been done

Reviewer 2 Report

The paper of Ben Jemâa and colleagues entitled "The maize caterpillar Mythimna (= Leucania) loreyi (Duponchel. 1827) (Lepidoptera: Noctuidae): Identification, distribution, population density and damage in Tunisia" reported data of the presence of Mythimna loreyi in Tunisia. The manuscript doesn’t sound as original scientific paper but as technical report. Although the authors monitored the pest throughout different sites and environmental conditions, no hypothesis has been made and I didn’t actually understand the aims of the work. Identification of specimens was based on morphological and molecular techniques already known, so that no original protocol has been proposed by the authors. The pest was monitored using Spodoptera frugiperda sex pheromone, but authors did not justify it. Wakamura et al. (2021) reported that pheromone-baited traps for S. frugiperda captured adult male of M. loreyi as well (https://doi.org/10.1007/s13355-020-00703-9), even though sex pheromone of M. loreyi has been already isolated (Takahashi et al. 1979, 1980). No specific information about the total number of traps used for monitoring was reported and results were reported in in a somewhat confused way. Moreover, why did authors report pairwise distances as a table? I believe that cluster diagram include all information needed. Finally, I suggest the authors to have the work linguistically revised. I’m sorry, but I suggest to reject the manuscript as the manuscript do not match sufficiently requirements to be considered as an original scientific paper.

Author Response

RESPONSE LETTER

Dear Editor,

We would like to thank you and the reviewers for the helpful feedback and verdict on our manuscript. In response to the comments of reviewer 2 responses are listed point by point herein.

Reviewer 2#:

1-The pest was monitored using Spodoptera frugiperda sex pheromone, but authors did not justify it. Wakamura et al. (2021) reported that pheromone-baited traps for S. frugiperda captured adult male of M. loreyi as well (https://doi.org/10.1007/s13355-020-00703-9), Even though sex pheromone of M. loreyi has been already isolated (Takahashi et al. 1979, 1980).

Response 1: Thank you for your comment. As part of the monitoring of the fall armyworm: surveys and sexual trapping network were carried out. The traps revealed the presence of another species of moth which was very abundant all along Tunisia. Thus, this work was conducted to explore its identification, distribution, damage and potential control program.

2-No specific information about the total number of traps used for monitoring was reported

Response 2: Thank you for your comment; in this work we have used one trap/0.5 ha. Thus, for Jendouba 120 and 82 traps. For Bizerte 12 in Mateur and similarly 12 in Utique. For Gabes 6 and for Nabeul 3 in each site.

3-Results were reported in in a somewhat confused way. Moreover, why did authors report pairwise distances as a table? I believe that cluster diagram include all information needed.

Response 3: Thank you for your comment, the table was presented in order to explain the pairwise distances of our samples and other species downloaded from NCBI. Additionally, the table could explain more the similarity between species.

4-Finally, I suggest the authors to have the work linguistically revised.

Response 4: Thank you for your comment, the paper was linguistically revised.

Reviewer 3 Report

Manuscript is well organized and is a good addition in insect fauna of Tunisia. Additionally, authors have done efforts with sound methodology to study population dynamics, incidence and seasonal occurrence of Mythimna (= Leucania) loreyi. Results will be helpful for future studies done for its IMP. However I have some suggestions for improvement in the attached file. Best of Luck

Author Response

RESPONSE LETTER

Dear Editor,

We would like to thank you and the reviewers for the helpful feedback and verdict on our manuscript. In response to the comments of reviewers we have made the suggested modifications and highlighted them in red color in the text. Detailed corrections are listed point by point herein.

Reviewer 1#: Comments for Authors Manuscript is well organized and is a good addition in insect fauna of Tunisia. Additionally authors have done efforts with sound methodology to study population dynamics, incidence and seasonal occurrence of Mythimna (= Leucania) loreyi. Results will be helpful for future studies done for its IMP. However I have some suggestions for improvement 

1-Abstract: This part addition of results in the form of numerical terms. This part looks very short. Add initially about the importance of insect you studied then its economic, little methodology and then results along with gaps too be studied. Results of identification missing in abstract. Please add your results here. 1. Line 26 add introduction of insect in two lines 2. Line 28-29 need rewrite up 3. Lines 31-31 needs numerical values of obtained results. 1. Introduction

Response 1: Thank you for your comment, the recommended modification has been done

Mythimna loreyi (Duponchel) is among the best-known noctuid moths in Africa and in other regions around the world. This pest is widely distributed in the Mediterranean regions. A survey was conducted during 2020 and 2021 to study the emerging lepidopteran pests inflicting cereal crops in Tunisia with specific emphasis on maize and sorghum. A dominant species was collected from traps placed all over the country. Thus, this study carried out first report on the identification, distribution, population density and damage of this pest. Results showed that Mythimna loreyi (Noctuidae) was the most abundant in Tunisia with total adult captures reaching 4779 and 9499 moths on sorghum and maize respectively during 2020. In addition, the outcomes of this study are discussed in terms of the wide distribution of this pest with food preference towards sorghum and maize, population dynamics and impact of weather parameters on its fluctuation in the field. This study provides important insights into the population biology of M. loreyi information that is required to establish an effective pest control program.”

2-Identification related information missing in this part. You may add also  1. line 38 please add author name and year with scientific name of species. line 62-63 please add more about population dynamics of this pest of related organisms. line 53-71 mostly you have addressed only host and biology of this pest. Kindly address the title of manuscript and write about its distribution, population density and damage in relevant studies, if not available then you may discuss about the closely related insect pest.

Response 2: Thank you for your constructives comments, the recommended modification has been done

“The maize caterpillar Mythimna (= Leucania) loreyi (Duponchel, 1827) (Noctuidae) is commonly called Loreyi leaf worm, the cosmopolitan rice armyworm, rice cutworm, cereal armyworm, or false armyworm [1-4]. This noctuid is a native species in East Asia [5]. It undergoes multiple generations annually [6, 7]. It is ranked among migratory Lepidopteran pests that reach the northern temperate zone [8]. This pest is reported on several gramineous hosts including 9 Poaceae species (Arundo donax, Avena sativa, Oryza sativa, Pennisetum purpureum, Saccharum officinarum, Sorghum bicolor, Triticum aestivum, Triticum durum and Zea mays), two Solanaceae species (Capsicum spp., Nicotiana tabacum) and one Fabaceae species Cicer arietinum [2,9-11]. M. loreyi has been emerged in Africa, Australia, the Middle East and Asia [12].

Kornosor [13] reported that M. loreyi infests particularly maize and reduces yields significantly when attacks occurred just before silking and pollination. Likewise, Qian et al (14) indicated that maize is the optimal host for M. loreyi leading to best biological performances (Larval growth duration 20.18 days, survival rate egg-pupa 68%, female pupa weight 32.4 mg). Earlier, Kornosor [14] cited that M. loreyi is a leaf feeder that invades maize from early growth stages (2 – 4 leaves) to pollen shedding and induced severe losses mainly on late-sown maize. Similarly, Sertkaya and Bayram [15] indicated that M. loreyi caused substantial economic damage to Z. mays. Additionally, Guo et al. [16] reported that M. loreyi usually occurred together with the closely related species Mythimna separata (Walker 1865) and caused considerable damage on host plants. Caterpillars bore into host plants and attacked the developing flower spikes. Moreover, Hirai et al. [17] cited that M. loreyi induced significant losses on crop production.

Mythimna loreyi attacks all parts of maize plants except the roots [18]. Newly hatched larvae feed on the tender leaves and make holes in their edges. As the larvae get older, they damage various parts of the plant as they feed on the developing tops and new growths of the leaves. They attack the stems and feed on grains in the milky stage. When they appear in large numbers, they caused economic damage to corn [19]. It is considered as one of important pests on maize crop due to its attacks that reduced the yield [20]. Whereas there is little information available concerning M. loreyi population dynamics in the field [17]. Insect population dynamics are subjected to various environmental factors, like temperature that is considered as a main factor influencing development, survival rate as well as fecundity of insects [3, 21, 22].

In this regard, previous research indicated that climatic parameters such as temperature, relative humidity, rainfall, and wind speed significantly influenced population dynamics of M. loreyi [18]. Furthermore, insects’ development need an optimum range, since, any high or low temperature could affect diversely development, reproduction, and survival [23]. According to previous research conducted by Qin et al. [3] M. loreyi can develop, survive, and lay eggs at temperatures ranged between 18 and 30 °C. Furthermore, temperature has greatest influence on insect multiple life-history variables, seasonality, insect activity and population dynamics in the field [3]. The adult insect most often comes out during the night period, especially at sunset. The adult is observed to emerge during the day and finds its way out of the cocoon through a letter-shaped slit. In the dorsal side of the cocoon, and immediately after its emergence, the adults move in different directions very slowly for about five minutes until their wings extend to their full size [16].

The number of M. loreyi generations varied from a country to another, for Korea one generation has been observed against two generations in Netherland, North America, and Costa Rica [24]. Besides, Salis et al. [25] indicated that insect density influenced significantly damage to the crop. Also, these authors pointed out that changes in population densities and outbreaks of insects are mainly affected by biotic and abiotic constraints. Furthermore, Wang et al. [26] showed that increases population density of M. separata caused greater damage to crop. Similarly, these authors demonstrated that study of insect densities could extend our knowledge of agricultural pests’ biology and assist to develop appropriate management strategies “

3-Materials and Methods

Line 81-82 you may give here ecology of these regions like annual temperature, rainfall etc. may be added with citation as these variables are important for pest distribution and dynamics.

Response: Thank you for your constructive comment, the recommended modification has been done as bellow:

“For more details, Medhioub et al. [29] indicated that in Jendouba region the average temperature varied from 5 to 10 °C and from 25 to 30 °C in winter and summer respectively. This region belongs to the humid climate with an annual rainfall of to 1000 mm per year. Additionally, these authors cited that Nabeul region is dominated by a temperate climate. The minimum temperature was 8.4 °C and maximum temperature was 15.8 °C in January, while, in August, minimum temperature reached 22.6 °C.

Gabes region is located at the Southern-East of Tunisia presented monthly average temperatures around 10 to 12 °C in winter against 30 °C in summer. Moreover, Jemai et al [30] reported that Gabes region had an arid Mediterranean climate, with mild wet winters and hot dry summers. Rainfall is typically scarce with annual mean varied between 100 and 200 mm [29]. For Bizerte region temperature varied between 7 °C as a minimum and 15 °C as a maximum and between 19 °C and 31 °C during winter and summer respectively. The annual average rainfall was between 510 and 638 mm [31].”

Line 88-95 methodology need citation

Response: Thank you for your constructive comment, the recommended modification has been done

Line 84-85 cannot understand why you gave table 1 describing GPS altitude etc. Either remove this information or relate it with relevant part in your text. 

Response : Thank you for your constructive comment, the recommended modification has been done

. Line 118 line space check

Response: Thank you for your constructive comment, the recommended modification has been done

Line 116-127 methodology need citation

Response: Thank you for your constructive comment, the recommended modification has been done

. Line 129-130 need citation

Response: Thank you for your constructive comment, the recommended modification has been done

. Line 177-183 need citation 8. Line 190-191 need rewrite up.

Response: Thank you for your constructive comment, the recommended modification has been done

4-Results

Line 188 Morphological identification of male or female please mention

Response: Thank you for your constructive comment, the recommended modification has been done

Line 19-194 please gives main identification characters of both male and female.

Response: Thank you for your constructive comment, the recommended modification has been done “Based on the description given by Rishi and Khokhar [46] morphological identification revealed that this species belongs to the order Lepidoptera; Noctuidae family and the genus of Mythimna. Adults measured about 15 - 20 mm (Fig. 2, A-B). This species was characterized with spherical head in shape and somewhat depressed dorso-ventrally with its sides roubded (Fig. 2, A-B). The antenna presented different segments (Fig. 2, C). This species presented brown wings with pale-brown veins (Fig. 2, D-E). In addition, legs are also brown (Fig.2, F).”

Line 234 you mentioned Figure 5 reported how a figure can report Please correct it accordingly

Response: Thank you for your constructive comment, the recommended modification has been done

Rewrite please line 234-235. 5. Line 246-247 please rewrite

Response: Thank you for your constructive comment, the recommended modification has been done

. Line 248-250 sentence is etalic. Please correct it

Response: Thank you for your constructive comment, the recommended modification has been done

Line 254-255 please illustrate both male and female and label the main identification characters (fig 2-3).

Response: Thank you for your constructive comment, the recommended modification has been done.

“(A) Female adult of Mythimna loreyi, (B) Male adult of Mythimna loreyi, (C) The Antenna with different segments, (D) Anterior wings pale-brown with small dark spot in the middle, (E) Posterior wings, (F) legs”

Line 258 table 2 please illustrate this table, its relevant text is missing in manuscript.

Response: Thank you for your constructive comment, the recommended modification has been done.

“Only the samples that amplified successfully and showed a band of the expected size (710 bp) were sequenced. Maximum likelihood inference resulted in our samples (5 and 7) (Figure 4). Table 2 reported the pairwise distances of our samples and other species downloaded from NCBI. Results revealed that clustering with M. loreyi with very close genetic distances.”

5-Discussion

This part looks too much short. Please add relevant mini reviews relating with your results. Every part of results must be discussed with relevant citations. 

Response: Thank you for your constructive comment, the recommended modification has been done.

“Noctuid species are considered among the most dangerous pests on maize [53]. The results revealed that M. loreyi caused substantial damage with 31.05 % of infestation rate during 2020 in Jendouba region. In this regard, previous work demonstrated that 100% of crop losses occurred if control measures have not taken at suitable time [54]. Furthermore, results showed that infestation caused by M. loreyi varied through the time, and differed from a region to another. The lowest infestation and pest incidence were recorded in Gabes region during September. In addition, results demonstrated that M. loreyi attacked both crops maize and sorghum and that the captures depended on the region and the crop. However, total numbers of captured adults in sorghum crops were less comparing to the captured adults on maize crops. In this context, Singh and Chaudhary [55] indicated that M. loreyi species had moderate performance on sorghum crops.”

General Comments As a whole manuscript is a significant addition in taxonomy and ecological studies of Mythimna (= Leucania) loreyi. With the addition of suggestions it would be better formatted for researchers in the relevant field.  

Round 2

Author Response

ANSWER LETTER

Dear Editor,

We would like to thank you and the reviewers for the helpful feedback and verdict on our manuscript. In response to the comments of reviewers we have made the suggested modifications and highlighted them in red color in the text. Detailed corrections are listed point by point herein.

Reviewer 1#

-Insert this sentence: “In central Mediterranean (Sicily) this noctuid flies from March to november [.....].”

-Thank you for your comment, the sentence has been added. “In central Mediterranean (Sicily) this noctuid moths occurred from March to November [25]. Besides, Salis et al. [26] indicated that insect density influenced significantly damage to the crop.”

- 10%, 5%, 52 °C, 15-20.

-Thank you for your comment, the recommended modifications have been done.

- Delete "The antenna presented different segments": the antennae of all lepidoptera species are multi-segmented. Replace with length or sexual dimorphism.

-Thank you for your comment, the recommended modifications has been done.

-Delete "with different segments" replace with The antenna of adult.... male?, female?

- Thank you for your comment, the recommended modifications has been done  as you recommended: “(A) Female adult of Mythimna loreyi, (B) Male adult of Mythimna loreyi, (C) The Antenna of male adult, (D) Anterior wings pale-brown with small dark spot in the middle, (E) Posterior wings, (F) legs.”

- Origin: Tunisia? Region?

-Thank you, done “Male Genitalia of Mythimna loreyi collected from Tunisia region.

-Add this reference (see pag 2): Russo, P.; Bella, S.; Parenzan, P. Contributo alla conoscenza dei Nottuidi della Sicilia (Lepidoptera, Noctuidae). Phytophaga 2002, XI, 11-85.

- Thank you, done.

Reviewer 2 Report

I've just finish to revise the work of Ben Jemâa and colleagues entitled "The maize caterpillar Mythimna (= Leucania) loreyi (Duponchel. 1827) (Lepidoptera: Noctuidae): Identification, distribution, population density and damage in Tunisia". This was my second revision as in the first one I suggested to reject the manuscript. Although authors tried to amended the text following suggestions, they failed to improve the quality of the manuscript. Besides they did not include information about sample size, the work was made with no specific scientifc hypothesis. The authors reported data on Mhytimna loreyi captured in a multi-year monitoring program for Spodoptera frugiperda, without reporting any data on S. frugiperda. As already stated in my previous revision, although Wakamura et al. (2021) reported that pheromone-baited traps for S. frugiperda captured adult male of M. loreyi as well (https://doi.org/10.1007/s13355-020-00703-9), authors must justify it. I'm really sorry, but I suggest toi reject the manuscript.

Attached the authors can find a PDF file with several suggestions and comments.

Author Response

ANSWER LETTER

Dear Editor,

We would like to thank you and the reviewers for the helpful feedback and verdict on our manuscript. In response to the comments of reviewers we have made the suggested modifications and highlighted them in red color in the text. Detailed corrections are listed point by point herein.

Reviewer 2#

-Entomo-fauna should be corrected as only M. loreyi data were reported.

- Thank you, done

-If M. loreyi is among the best-known noctuid moths in Africa, how can this study be the "first report on the identification, distribution, population density and damage of this pest"?

-Thank you for your comment. This work carried out first extensive investigation of this pest in Tunisia. So, we considered this study as first report on the identification, distribution, population density and damage of this pest in our country.

- Dominant species doesn't sound correct. The term "dominant species" is associated to a species that is the most abundant in a community and affects the distribution and the occurence of other species. However, the analysis of species composition at community level should be performed to evluated that. Unfortunately, your data are not sufficient to study Lepidoptera community as you considered specific baited-trap for monitoring.

-Thank you for this comment. The term dominant species was omitted since the work has not been carried on Lepidoptera community.

- First of all, the scientific name of the pest has been already reported in full, so that here and all over the abstract must be reported as M. loreyi. Secondly, how can authors state that this species was the most abunt in Tunisia? Did the authors monitor different environments using several sampling procedures?

- Thank you for this comment. First, M. loreyi has been modified in the abstract. Secondly, the statement of M. loreyi as the most abundant in Tunisia was revised and corrected accordingly.

-This is valid for almost all insect species. I suggest to delete it.

-- Thank you, done

- Although temperature, rainfall amnd other climatic parameters could influence M. loreyi population dynamics, the work of Hong et al. (2022) did not prove that. Hong et al. (2022) evaluated exclusively the trapping efficacy of different sex pheromones towards S. frugiperda and M. loreyi, and did not perform any experimental trials to demonstrate the effect of climatic parameters on the population dynamics of these species. As authors know, climatic parameters can influence the release rate of pheromones, which consequently affect the number of adults in traps. I'm sorry to say that the sentence reported here is quite speculative.

- Thank you for your comment, the sentence has been corrected.

- What does it mean? As far as I concerned, I believe that it is implied that insect density affect damage to crop, as well as that populations are influenced to biotic and abiotic factors.

-Thank you. This sentence has been reported as a bibliography related to this work.

- Although authors replied to my question about sampling size, they did not include information in M&Ms section. How many hectares were monitored? How many traps? What about samplings?

- Thank you. Size of each field and the number of traps had been reported in Table 1. Concerning sampling, it was achieved weekly and randomly in each field.

- Standard deviation or standard error? I believe that it indicates SE.

- Thank you for this comment, yes it is SE.

- Are there some statistics (e.g., F value) that indicate this dependence? I think that authors stated this based on correlation analysis. However, how did authors perform correlation analysis?

- correlation analysis was performed using the test “r” Bravais Pearson.

Reviewer 3 Report

I have completed the review. Authors have improved the manuscript, but still I have noted some previous suggestions have not been included. Please add them carefully; these additions would make your manuscript valuable.

Author Response

ANSWER LETTER

Dear Editor,

We would like to thank you and the reviewers for the helpful feedback and verdict on our manuscript. In response to the comments of reviewers we have made the suggested modifications and highlighted them in red color in the text. Detailed corrections are listed point by point herein.

Reviewer 3#

I have completed the review. Authors have improved the manuscript but still I have noted some previous suggestions have not been included. Please add them carefully; these additions would make your manuscript valuable.

Abstract

-Please add results in numerical values related with distribution, population density and damage in Tunisia from your results in this part. They are still missing in your abstract.

Thank you for this constructive comment, the recommended modification has been done “Moreover, Infestation caused by M. loreyi varied from a region and a period to another. The mean infestation percentage reached its maximum during August at 31.05% and 20.69% for Jendouba and Bizerte respectively. While the highest infestations were observed in Gabes and Nabeul regions during July.”

Results

-Under heading (3.1. Morphological identification) I proposed the addition of main identification characters of male and female. The genitalia of both male and female along with illustration (labeled figures in figure 2-3) have not been given yet.

- Thank you for your constructive comment, the recommended modification has been done “Based on the description given by Rishi and Khokhar [46] morphological identification revealed that this species belongs to the order Lepidoptera; Noctuidae family and the genus of Mythimna: Female adult (♀) measured about 14 mm (Fig. 2, A) against 16.3 mm for male adult(♂) (Fig. 2, B).”

 Discussion

-This part is still short. 

- Thank you for your constructive comment, the discussion has been extended.